# Therapeutical Management and Drug Safety in Mitochondrial Diseases—Update 2020

**DOI:** 10.3390/jcm10010094

**Published:** 2020-12-29

**Authors:** Francesco Gruosso, Vincenzo Montano, Costanza Simoncini, Gabriele Siciliano, Michelangelo Mancuso

**Affiliations:** Department of Clinical and Experimental Medicine, Neurological Clinic, University of Pisa, 56126 Pisa, Italy; francesco.gruosso93@gmail.com (F.G.); v.montano89@gmail.com (V.M.); costanza.simoncini85@gmail.com (C.S.); gabriele.siciliano@unipi.it (G.S.)

**Keywords:** mitochondrial diseases, therapies, mitochondrial toxicity

## Abstract

Mitochondrial diseases (MDs) are a group of genetic disorders that may manifest with vast clinical heterogeneity in childhood or adulthood. These diseases are characterized by dysfunctional mitochondria and oxidative phosphorylation deficiency. Patients are usually treated with supportive and symptomatic therapies due to the absence of a specific disease-modifying therapy. Management of patients with MDs is based on different therapeutical strategies, particularly the early treatment of organ-specific complications and the avoidance of catabolic stressors or toxic medication. In this review, we discuss the therapeutic management of MDs, supported by a revision of the literature, and provide an overview of the drugs that should be either avoided or carefully used both for the specific treatment of MDs and for the management of comorbidities these subjects may manifest. We finally discuss the latest therapies approved for the management of MDs and some ongoing clinical trials.

## 1. Introduction on Mitochondrial Diseases

Mitochondrial diseases (MDs) are a group of genetic disorders characterized by dysfunctional mitochondria. Encompassing all pathogenic mitochondrial and nuclear DNA mutations, they represent the most frequent group of metabolic disorders in humans, with a prevalence of about 1 in 4300 cases [1]. Mitochondrial DNA (mtDNA)-related disorders, unlike other genetic disorders, are characterized by maternal inheritance. As such, mtDNA point mutations will be passed on by a mother to all her children (males as well as females), although only her daughters will be able to continue to transmit the mutation [2]. These diseases are clinically heterogeneous; this may be partly attributed to the heteroplasmy level of mtDNA molecules in cells and the threshold effect: mtDNA molecules are distributed in multiple copies in each cell (polyplasmy) but most pathogenic mutations do not usually affect all mtDNA copies (heteroplasmy). Additionally, there is a level to which the cell can tolerate damage to mtDNA molecules: metabolic dysfunction and clinical symptoms occur only when the mutation load exceeds this threshold [3]. Furthermore, there are limited genotype–phenotype correlations to direct molecular genetic diagnosis, and many phenotypes can be caused by defects involving numerous genes. For example, Leigh syndrome, the most common clinical phenotype seen in pediatric MDs, is a progressive neurodegenerative disorder which may be caused by mutations in almost 80 different genes [4].

MDs may manifest in childhood or adulthood, with vast clinical heterogeneity. Patients may show symptoms affecting a single organ or tissue, or multisystem involvement; the most affected organs are usually highly dependent on aerobic metabolism and the disease is often progressive, with high morbidity and mortality [5]. Some of the clinical features shown by patients may indicate specific syndromes. MDs should be suspected when an individual presents with the clinical involvement of more than one tissue and/or organ, especially when both the central and peripheral nervous system are affected.

Mitochondrial dysfunction may also be found in neurodegenerative disorders that are not classified as “primary mitochondrial diseases”, such as Alzheimer´s disease, in which the role of mitochondria is one of the major factors in disease progression [6,7]. As a result, it is clear why studies focused on the pathophysiological mechanisms underlying MDs are becoming increasingly important in medical research. 

Because of the clinical heterogeneity of MDs, diagnosis may be challenging, especially considering that the patient’s phenotype may overlap with a broad range of diseases. MDs, often suspected in early childhood, may be diagnosed thanks to proper investigation with regard to clinical presentation, family history, pathology, metabolic profiling, enzyme activity levels, and the use of specific techniques such as electrophysiology, magnetic resonance imaging (MRI), magnetic resonance spectroscopy (MRS), and genetic analysis [8].

For the vast majority of these diseases, therapy is only symptomatic, although several recent clinical trials have highlighted the value of disease-modifying therapies such as idebenone [9,10,11] or adeno-associated viral vectors [12] in the treatment of Leber’s hereditary optic neuropathy (LHON) or the supplementation of coenzyme Q10 (CoQ10) in patients with CoQ10 deficiency [13]. Management of patients includes the prevention and treatment of early complications that may affect the involved organs, avoiding potential triggers of decompensation such as fasting, intercurrent illness, pyrexia, trauma, surgery, or use of medications toxic to mitochondria [14]. Furthermore, several studies in the literature suggest the importance of non-pharmacological treatment as an additional therapeutic strategy: an example is the prescription of a ketogenic diet in patients suffering from metabolic or neurological diseases, including MDs. In fact, thanks to the greater supply of ketones, the brain should have a more efficient energy source than that provided by glucose alone [15].

Currently, the guidelines for the correct diagnosis and therapy of diseases undergo constant and periodic assessment in all fields of medicine. A complete review of therapeutic management of mitochondrial diseases and drugs that should be used or avoided in mitochondrial patients is, therefore, particularly important. Indeed, due to the pathophysiological processes underlying them, MDs represent a distinct category in the clinical panorama and deserve specific analysis.

## 2. Safety of Drug Use in MDs

As discussed above, the extreme variability in the phenotype and genotype of MDs suggests the need to identify a specific therapy for every single patient under examination. Subsequently, the vast number of drugs available in clinical practice should be analyzed in order to separate drugs that may be defined as “safe” from those that may instead be dangerous in the treatment of the patients affected by a mitochondrial disease. For instance, there are some categories of antibiotics which have effects on mitochondrial translation and therefore may be dangerous for patients harboring mitochondrial translation deficiency, as demonstrated by Jones et al. in their study [16].

For this reason, an international group of experts proposed a study aimed at developing a consensus on safe medication use in patients with a primary mitochondrial disease [14]. The authors highlighted how, for most existing licensed drugs, mitochondrial toxicity in vivo is unknown. In the evaluation of drug safety in subjects affected by a mitochondrial disease, the only available information is derived from in vitro and in vivo pre-clinical studies and published case reports. Thus, a consensus based on a systematic analysis of notions present in literature and the clinical experiences of pediatricians, internists, and neurologists qualified in treating this type of patient may certainly be useful for strengthening knowledge in the treatment of patients affected by these particular diseases. 

In this workshop, using a modified Delphi-based technique, a group of internationally acknowledged experts was able to develop a consensus on drug safety in the treatment of patients with MDs. The Delphi method, developed by the RAND (Research and Development) Corporation in 1953, is a consensus method used in research that is directed at problem-solving, idea generation, or for determining priorities [17]. The technique is based on a structured and repetitive survey process of at least two rounds, which continues until a consensus is reached among panelists. Between each round, feedback is provided to the panelists. Due to the short duration of the workshop (2 days), only a limited number of drugs were considered. Thus, after selecting the drugs to study as per a previously published list on the website of the patient advocacy group IMP (https://www.mitopatients.org/mitodisease/potentiallyharmful-drugs), supplemented with the names of a few drugs commonly prescribed to patients affected by MDs (i.e., anesthetic agents, analgesics, antibiotics, and antiepileptic drugs), the total number of drugs/drug classes was limited to 46 [14].

The expert panel reported that all the 46 drugs or drug groups studied were generally safe for patients with MDs. In particular, they reported a good or strong consensus for six drugs or drug groups in the first Delphi round: enalapril, paracetamol, midazolam, carbamazepine, oxcarbazepine, and haloperidol. For the other 40 drugs analyzed, a further examination was necessary based on the clinical experiences of the panelists involved. At the end of this workshop, the experts identified some specific restrictions that should be evaluated in relation to certain molecular alterations and particular clinical conditions.

The main recommendations proposed at the end of this workshop are summarized in the following points:

(1) The use of aminoglycosides for elective long-term treatment should be preceded by a screening for the homoplasmic m.1555A>G and m.1494C>T mutations associated with both aminoglycoside-induced and non-syndromic hearing loss [18]. Exceptionally, when an effective broad-spectrum antibiotic treatment is needed in emergency situations and the prescription of aminoglycosides is considered necessary, this drug classes may be administered until the immediate danger has passed or antibiogram has been carried out. After that, physicians should replace aminoglycoside with a safer antibiotic class. 

(2) Valproic acid should be administrated only in exceptional situations. *POLG* mutations are an absolute contraindication for use of this drug. In fact, in the case of clinical signs suspicious for *POLG* disease (i.e., epilepsia partialis continua, explosive onset of focal epilepsy, or rhythmic high amplitude delta with superimposed spikes on electroencephalogram (EEG) and/or known liver disease, valproic acid should not be used in patients [19].

(3) Neuromuscular blocking agents should be carefully used and monitored for patients manifesting a mainly myopathic phenotype. General anesthesia has not shown side effects in patients with MDs, and experts agreed on the safety of these drugs and drug classes, while recommending care during general surgical procedures for these patients. A good strategy to prevent the effects of catabolism may be to minimize preoperative fasting and to administer intravenous glucose perioperatively in the case of prolonged anesthesia unless the patient is on a ketogenic diet. 

(4) Adverse effects may be influenced by the duration of drug administration, which should be guided by individual patient necessities and their response to specific treatments. Indeed, there may be clinical circumstances in which subjects benefit from the longer administration of some drugs, although this involves an increased risk of side effects or disease progression, particularly when better alternative pharmacological options are not available. 

(5) Renal impairment is a common feature of many patients, and therefore drug dose adjustment should be considered, especially for drugs with a predominantly renal clearance. Due to the major risk of developing metabolic acidosis (lactic acidosis), physicians should be careful in administering the drugs that most frequently cause acidosis, performing regular clinical reviews and monitoring acid–base status in blood. 

The main limitation of this type of study is that expert judgement is not always based on empirical studies, although the consensus model represents the most-used method to inform clinical practice in the absence of validated data. Furthermore, the physicians´ experience that patients seemed to tolerate most drugs is not strongly supported by pre-clinical studies. This may be partially attributed to the use of different dosages in pre-clinical studies and clinical practice, where the reference dose for drugs is much lower than the toxic concentration dose. Conversely, higher or toxic drug levels are used in the majority of pre-clinical studies.

Another important limitation of this study is that patients often have co-morbidities and/or are taking more than one drug; this implies a greater difficulty or an impossibility in determining which effects are due to the drug in question.

For these reasons, it is important to update the current list regularly, considering the addition of other drugs in the future using the same Delphi process. Furthermore, due to the frequent discrepancies between pre-clinical and clinical situations, future pre-clinical studies should assess the toxicity of drug with an analysis closer to the actual doses used in the treatment of patients and their conditions (e.g., mitochondrial deficiencies), in order to improve knowledge on drug-induced mitochondrial dysfunction in these subjects. According to the results of this consensus paper and the clinical trials in literature, we proposed a review of recommended therapeutic management of primary mitochondrial diseases, distinguishing the management of patients affected by peripheral neuropathies and skeletal muscle diseases from that of patients in which the disease mainly affects the central system. A list of drugs analyzed in this paper is summarized in the Table 1.

## 3. Management of Myopathy and Neuropathy: Therapeutic Strategies in Skeletal Muscle and Peripheral Neuron Involvement

### 3.1. Myopathy

Damage to the cell respiratory chain, caused by mutations in mitochondrial or nuclear genes encoding enzymes involved in oxidative phosphorylation, is the main mechanism clinical manifestations of MDs are based on. The consequent reduction of ATP molecules preferentially affects organs with high energy requirements such as skeletal muscle. Myopathy, expressed by symptoms of exercise intolerance with premature fatigue or muscle weakness, is therefore common in MDs, for which there is no effective available treatment [47]. Myopathy may represent the only clinical feature of mitochondrial diseases, as observed in some patients affected by primary mitochondrial myopathy (PMM), or, more commonly, patients may show additional manifestations (i.e., diabetes, sensorineural hearing loss, optic atrophy, peripheral neuropathy, cardiomyopathy, nephropathy, hepatopathy, stroke-like episodes, seizures, ataxia, failure to thrive, developmental delay or regression, and dementia) [38,48].

Anamnesis and physical examination should be carefully performed to identify the locus of the impairment. Exercise intolerance may involve functions related to the lungs (i.e., obstructive pulmonary disease), heart (i.e., cardiomyopathy), or the motor unit (i.e., primary disorders of nerve or muscle). In contrast, the impairment may be due to an insufficient amount of ATP in working muscles, which may be observed, for example, in patients affected by inborn errors of intermediary metabolism (i.e., myophosphorylase deficiency or carnitine palmitoyl transferase deficiency) or mitochondrial myopathies [39]. 

For these reasons, studies focused on exercise tolerance and physical training have been conducted to analyze the effect of exercise in patients affected by mitochondrial myopathy. The use of the 6-minute walking test (6MWT) as a functional test in clinical research may be associated with laboratory and physiological measures (i.e., resting and end-exercise blood lactate, respiratory exchange ratio (VCO_2_/VO_2_), measurement of cardiac output, and pulmonary function measured with spirometry), in order to identify patients affected by PMM [48]. In addition, because of its sensitivity to fatigue-related changes, the 6MWT may represent a measure of fatigability [49].

Jeppesen et al. reported how aerobic training for 12 weeks significantly improved maximal oxidative capacity (VO_2max_) in 20 persons with four different mtDNA mutation types and a variety of mutant loads. Furthermore, they reported how this improvement translated into clinically significative effects only in subjects with severe oxidative defects and not in asymptomatic patients harboring mtDNA, in which daily activities did not change. As reported, there were no changes in mtDNA mutation load in muscle, plasma creatine kinase (CK) levels, and muscle regeneration and apoptosis with physical training. Short-term exercise seems to be safe in these patients [31]. Thus, we may assume that regular aerobic exercise should be included in a multi-disciplinary approach to these patients. 

Although long-term efficacy has not been definitively confirmed, physicians usually prescribe dietary supplements like antioxidants and mitochondrial cofactors. Complex B vitamins (mainly thiamine/vitamin B, and riboflavin/vitamin B2), creatine, coenzyme Q10 and its reduced form ubiquinol, alpha lipoic acid, N-acetylcysteine, folinic acid, vitamin C, and vitamin E may be included among the compounds of these supplements [32], which are often described as “mito-cocktails”.

The evidence supporting the use of coenzyme Q10, also known as ubiquinone, in mitochondrial disease, based on Level III and IV open-label studies, was revised in 2007. The main limitations of these studies are the low dosages of CoQ10 and the lack of information on blood or tissue levels [33]. Reduced CoQ10 has become commercially available in the form of ubiquinol, which has a higher bioavailability (it is three to five times better absorbed when compared with the oxidized form of CoQ10, ubiquinone). Ubiquinol is administrated at doses of 2 to 8 mg/kg per day, while ubiquinone is usually prescribed at doses of 5 to 30 mg/kg per day. Both supplements are administered twice daily with meals [34].

Riboflavin (vitamin B2) serves as a flavoprotein precursor and so represents an essential element in complexes I and II of the mitochondrial respiratory chain and a cofactor in enzymatic reactions such as the fatty acid β-oxidation and the Krebs cycle. Consequently, riboflavin supplementation may improve symptoms and the clinical course of subjects affected by multiple acyl-CoA dehydrogenase deficiency (typically caused by electron-transport flavoprotein dehydrogenase (*ETFDH)* gene mutations) [40], mitochondrial diseases with complexes I and II deficiencies (as reported by some non-randomized studies in the literature) [50,51], and acyl-CoA dehydrogenase-9 (*ACAD9*) deficiency (which results in increased complex I activity in fibroblasts of patients). The usual prescribed dosage of riboflavin is 50–200 mg/day, divided into 2–3 doses [41].

Creatine, in its phosphorylated form (phosphocreatine), serves as a source of high-energy phosphate released during anaerobic metabolism. The highest concentrations of creatine are observed in skeletal muscle and the brain, which are tissues with higher energy requirements. Lower phosphocreatine levels may be observed in skeletal muscle and the brain, respectively, in individuals with mitochondrial myopathies and mitochondrial encephalomyopathies [52,53].

Treatment based on creatine supplements results in an increase in high-intensity, isometric, anaerobic, and aerobic power, as shown in some studies reported in the literature [28,54]. The standard dosage that is actually suggested is 10 g per day for adults twice daily, and 0.1 g/kg per day, divided into two doses for children.

l-Carnitine is a cellular compound involved in the process of mitochondrial β-oxidation of fatty acids and the esterification of free fatty acids that may otherwise be sequestered by CoA. Tissues like those of skeletal muscle, heart, and liver mostly depend on β-oxidation for ATP production. Carnitine may prevent CoA depletion and remove excess, potentially toxic, acyl compounds. It is possible observe, in individuals with respiratory chain defects, a reduction of free carnitine levels in plasma and an increase in esterified carnitine levels, even if primary carnitine deficiency is not a typical feature of MDs. l-Carnitine supplementation for mitochondrial disorders may contribute to restoring free carnitine levels and remove accumulating toxic acyl complexes [34]. Carnitine may be administered orally or intravenously in mitochondrial disease and is usually combined with other vitamins and cofactors. Currently, the benefit of isolated use of carnitine in patients with primary mitochondrial disorders has not been confirmed, as for other supplements used separately or as a part of an antioxidant cocktail in the treatment of mitochondrial patients (i.e., thiamine (B1), vitamins C and E, and alpha-lipoic acid). 

One small, randomized placebo-controlled trial provided evidence for the combination of creatine, CoQ10, and lipoic acid [29]. Patients usually take dietary supplements when recommended by a physician or on their own. In this case, serious side-effects were not reported, and some subjects reported an improvement in their activities [37].

With regard to the use of drugs, it has also been suggested that care should be taken when using agents such as statins, corticosteroids, metformin, and antiretrovirals, since they may worsen the underlying myopathy [27]. For example, statin-induced myopathies are associated with the inhibition of the Qo site of respiratory complex III (CIII) by several statin lactones. Consequently, polymorphisms of uridine 5′-diphospho-glucuronosyltransferases (UGTs), the enzymes converting statin acids into lactones, and CIII could be predisposing factors in statin-induced myopathies [43]. 

### 3.2. Neuropathy

The prevalence of peripheral neuropathy in MDs is about 12.4%. Mitochondrial neuropathic patients have an increased prevalence of ataxia, hearing loss, muscle weakness and muscle wasting [55]. Individuals harboring *POLG*-, *TYMP*-, or *MPV17-* deletions or m.8993T>G and m.8993T>C mtDNA mutations, as well as those affected by *SURF1*-related mitochondrial diseases are at greater risk of developing peripheral neuropathy. This may also be a secondary complication of mitochondrial diabetes, renal insufficiency, or a side effect of treatment [27].

Patients with “pure forms” of neuropathy are rare, instead, it is more frequent to observe a mixed/undefined pattern. Because of the relative rarity of the pure forms, it is very difficult to find a strict genotype–phenotype relation. Neuropathic pain seems to be more frequent in *POLG* patients than in mitochondrial patients with neuropathy with different genetic causes [55].

The use of some drugs should be avoided in these patients, in particular antiviral drugs and dichloroacetate. Patients treated with nucleoside analogue reverse transcriptase inhibitors (NRTIs) may develop myopathy or neuropathy after long-term therapy. Zalcitabine, didanosine, and lamuvidine cause neuropathy, zidovudine causes myopathy, and stavudine and fialuridine cause neuropathy or myopathy and lactic acidosis. Muscle wasting, myalgia, fatigue, weakness, and CK elevation represent the main features of myopathy; indeed, the neuropathy is painful, sensory, and axonal. Another cause of NRTI-related neuropathy is mitochondrial toxicity. Thus, NRTI-induced mitochondrial dysfunction influences the clinical administration of these agents, especially at high doses and when combined [24].

Particular attention should be paid to treatment with dichloroacetate (DCA), especially in patients affected by mitochondrial myopathy, encephalopathy, lactic acidosis, and stroke-like episodes (MELAS). Dichloroacetate (DCA) has been used to treat congenital and acquired conditions associated with lactic acidosis, thanks to its ability to reduce lactate. In fact, DCA is able to interact with the pyruvate dehydrogenase enzyme complex, located in the mitochondria [30]. Because of the increase in pyruvate catabolism, the accumulation of lactate is prevented, and lactic acidosis may be avoided in many patients. Different studies demonstrated that dichloroacetate treatment was well tolerated and blunted the postprandial lactate increase in children with congenital lactic acidosis, although neurological or other measures of clinical outcome improvements were not observed [20,21]. However, in individuals with MELAS syndrome it was demonstrated that treatment with dichloroacetate was linked to peripheral nerve toxicity. This may be explained by the greater vulnerability of MELAS m.3243A>G patients to DCA toxicity as compared to patients with other diseases causing lactic acidosis. Furthermore, diabetes mellitus is likely an additional contributing factor, given that it commonly causes symptomatic or subclinical neuropathy with both axonal and demyelinating features [30].

### 3.3. New Potential Primary Mitochondrial Myopathy (PMM) Treatment

In recent years, different clinical trials on potential treatment for PMM have been started and are still ongoing to analyze the effect of new molecules that may provide benefits to these patients.

One of these molecules, elamipretide, demonstrated a benefit measured by an improvement of exercise performance after 5 days of treatment in patients with PMM without increased safety concerns [44]. Thanks to selective binding to cardiolipin via electrostatic and hydrophobic interactions, elamipretide can protect from oxidation and, consequently, preserve mitochondrial cristae, promote oxidative phosphorylation, and inhibit mitochondrial permeability (transition pore opening). Kaara et al., in a randomized, placebo-controlled clinical trial of elamipretide in patients with PMM, observed a dose-dependent improvement in 6MWT results, which confirmed the benefits recorded in exercise performance in aged mice reported in preclinical animal studies [45]. The trial also found a correlation, evaluated for several cardiopulmonary exercise testing (CPET) parameters, between the change in distance walked in 6MWT and peak oxygen consumption in all the participants, similar to what has been observed in other advanced chronic conditions such as heart disease [44]. Although the trial provided Class I evidence, the improvements in exercise performance and the well-tolerated safety profile of elamipretide are encouraging. 

Omaveloxolone is another molecule under investigation that may be potentially useful in treatment of PMM. It is a semi-synthetic oleanolic triterpenoid and a potent activator of nuclear factor erythroid 2-related factor 2 (Nrf2). Triterpenoids are a class of small anti-inflammatory molecules derived from natural sources. Omaveloxolone targets redox-sensitive cysteine residues on the regulatory molecule Keap1 and thereby rescues Nrf2 from degradation. Moreover, blockage of Keap1 inhibits the NF-κB proinflammatory signaling pathway. These effects of omaveloxolone may improve muscle function, oxidative phosphorylation, antioxidant capacity, and mitochondrial biogenesis in patients with mitochondrial myopathy [46]. The clinical trial is still ongoing, but it has already shown that treatment with 160 mg omaveloxolone leads to a significant reduction in lactate production and heart rate during submaximal exercise. As most everyday activities are of a submaximal intensity, these results are potentially clinically meaningful and indicate that omaveloxolone may improve exercise tolerance in patients with mitochondrial myopathy [46]. The study provided Class II evidence that, for patients with mitochondrial myopathy, omaveloxolone compared to placebo did not significantly change peak exercise workload.

Thymidine kinase 2 (TK2) deficiency may determinate a late onset case of mild chronic progressive external ophthalmoplegia (CPEO). The most frequent clinical presentations are infantile onset and childhood onset progressive limb and bulbar myopathy with restrictive lung disease. Domínguez-González et al. performed a clinical trial based on the use of pyrimidine deoxynucleoside and deoxynucleotides as novel pharmacological therapies in 16 patients with mitochondrial myopathy due to TK2 deficiency. The beneficial effects of the therapy were confirmed both with functional tests, such as 6MWT, and laboratory measures. Serum levels of growth differentiation factor (GDF-15), a sensitive diagnostic biomarker for mitochondrial myopathy, were reduced in these subjects. The benefits observed, including stabilization or mild improvements in motor and respiratory functions, were better in early-onset patients then in late-onset ones [42]. Further studies are ongoing to support potential of this treatment.

## 4. Management and Treatment of Patients with Central Nervous System Involvement

### 4.1. Seizures and Epilepsy 

Epilepsy is a major feature of mitochondrial disease. The main genetic causes of mitochondrial epilepsy are: mtDNA mutations (including those typically associated with MELAS and myoclonic epilepsy with ragged red fibers (MERRF) syndromes); mutations in *POLG* (classically associated with Alpers syndrome but also presenting as mitochondrial recessive ataxia syndrome (MIRAS), spinocerebellar ataxia with epilepsy (SCAE), or myoclonus, epilepsy, myopathy, sensory ataxia (MEMSA) syndromes) and other disorders of mitochondrial DNA maintenance; complex I deficiency; disorders of coenzyme Q10 biosynthesis; and disorders of mitochondrial translation such as *RARS2* mutations [56]. Epileptic seizures may occur sporadically or as a part of a complex syndromic disorder, such as those mentioned above. The management of mitochondrial epilepsy may be very complicated, and prognosis may be often negative. With regard to the treatment, multiple anticonvulsants are frequently prescribed to individuals with mitochondrial epilepsy. There are no studies that clearly prove the role of vitamins or other nutritional supplements [56]. A recent multicenter Italian study collected data on a large number of patients affected by mitochondrial epilepsy and their management [22]. The authors reported a full or partial response to antiepileptic drugs (AEDs) for the most of patients, with an absence of response in only 11% of cases. Response to treatment, however, does not seem to be influenced by age at epilepsy onset. In clinical practice, mitochondrial epilepsy is generally treated using AEDs characterized by low mitochondrial toxic potential, for example gabapentin, lamotrigine, zonisamide, and levetiracetam (LEV). LEV was the most-used AED in the Italian cohort (mainly in adults), with an effectiveness in 81% of the adult-onset epilepsy cases. Instead, in early-onset epilepsy, the authors reported a more frequent use of phenobarbital and vigabatrin, achieving seizure control in 75% and 66% of cases, respectively. It is interesting to highlight how the use of topiramate (TPM) in children allowed full seizure control, although this drug was used in only 25% of cases.

The role of adrenocorticotropic hormones and a ketogenic diet in the treatment of these patients has been reported by some studies in the literature [57,58] which highlighted how these options are effective and safe. However, the Italian study reported very poor antiepileptic efficacy in young children.

As reported above, valproic acid should be avoided, particularly in patients with *POLG* mutations, because this drug may cause a fulminant hepatic failure or worsen neurologic symptoms, including seizures. 

Epilepsy may be refractory to treatment. No specific combination of medications has been observed that results in clinical improvement. 

### 4.2. Stroke-Like Episodes

Stroke-like episodes may be the main feature of some forms of MDs, especially in MELAS syndrome [59]. The more common features of these episodes include headache, nausea and vomiting, encephalopathy, focal-onset seizures (with or without associated focal neurological deficits), and cortical and sub-cortical signal abnormalities not confined to vascular territories, known as “stroke-like lesions”. While originally described in young adult patients, this type of symptom has increasingly shown late-onset forms [60]. 

The exact pathophysiological mechanisms of stroke-like episodes are unknown, and this is also reflected in debates on the management of stroke-like episodes. For example, the use of l-arginine seems to be supported by vascular involvement in stroke-like episodes and the nitric oxide deficiency found in patients [36]. However, there is no strong evidence that l-arginine (associated or not to co-supplementation of citrulline) improves the management of stroke-like episodes, so further investigations are needed.

In the case of clinical suspicion of a stroke-like episode, a detailed history of patient is a diagnostic key point. Indeed, a sudden onset of focal neurological deficits, particularly pure motor weakness (facial weakness or hemiparesis), that evolve rapidly within minutes should direct the physician towards a vascular stroke, since these modalities of symptoms onset are quite unusual in stroke-like episodes. Instead, for any individual who presents with complex visual symptoms, perception problems, and hearing disturbances persisting for hours or days before admission, the hypothesis of a mitochondrial stroke-like episode is more likely. It is also important to try to identify potential triggers such as infection, gut dysmotility, dehydration, prolonged fasting, and non-adherence to the anti-epileptic drug(s) [23]. 

Patients who have already had a stroke-like episode and show symptoms suggestive of a new episode or seizure should be considered for early treatment with benzodiazepines if they are outside the hospital, and with an intravenous anti-epileptic drug (AED) such as levetiracetam, phenytoin, lacosamide, or phenobarbitone once in the hospital environment. As mentioned above, the use of valproate is not recommended for patients harboring *POLG* mutations [19]. 

When intensive care is necessary (i.e., generalized, convulsive status epilepticus; intrusive, frequent focal motor seizures with breakthrough generalized seizures which fail to respond to intravenous AEDs; severe encephalopathy with a high risk of aspiration; or focal motor status epilepticus with retained consciousness failing to respond to benzodiazepine and to intravenous AEDs) the use of midazolam is recommended as general first option for anesthesia. With regard to the use of propofol there is no strong evidence suggesting it should be totally avoided in refractory status epilepticus associated with stroke-like episodes. Therefore, physicians should evaluate its administration case by case [23].

For patients who show neuro-psychiatric symptoms, the use of haloperidol, benzodiazepine, and quetiapine is suggested by clinical experience as safe and efficacious, according to the most recent recommendations published after a recent consensus on stroke-like episode management [23]. 

### 4.3. Hearing Loss

Hearing loss is a common feature of MDs. Patients most frequently show progressive sensorineural hearing loss, typically of cochlear origin. Alternative observed forms are congenital hearing loss and auditory neuropathy [27]. Mutations in mtDNA are associated with both syndromic and non-syndromic hearing loss. In particular, m.1555A>G and m.3243A>G mutations are frequently associated with sensorineural hearing loss. Regarding syndromic hearing loss, systemic neuromuscular disorders such as MELAS, MERRF or PEO syndromes are often characterized by sensorineural hearing loss. *MT-RNR1* or the mitochondrially encoded tRNA serine 1 (*MT-TS1*) gene may show many pathogenic variants associated with non-syndromic hearing loss, which is generally symmetric and progressive, mainly involving a high frequency range. m.1555A>G in *MT-RNR1* is the most common non-syndromic mutation [61]. 

Digital hearing aids may improve function in the case of moderate or severe hearing loss, whereas cochlear implantation should be considered in profound hearing loss. In the management of mitochondrial diseases, aminoglycosides may precipitate the development of hearing loss and should be used with caution, as should platinum-based anticancer drugs [26]. Aminoglycoside and cisplatin ototoxicities are also related to the induction of oxidative stress and an increase in reactive oxygen species (ROS) levels that may exacerbate mitochondrial disease. Consequently, mitochondria-targeted antioxidants could be useful for the prevention and treatment of diseases associated with mitochondrial dysfunction. Examples of these drugs include lipophilic cation-based antioxidants such as MitoQ, MitoVitE, MitoPBN, MitoPeroxidase, SkQ1, and SkQR1, or amino acid- and peptide-based antioxidants such as SS tetrapeptides [62,63].

However, there is a lack of clinical evidence for the use of mitochondria-targeted antioxidants, although there are reported animal studies in the literature [64]. Therefore, clinical trials based on mitochondria-targeted antioxidant administration need to be performed to confirm the otoprotective effects in patients.

### 4.4. Visual Loss

Ophthalmologic manifestations in MDs are present in 35–81% of cases and have been described in many forms of mitochondrial disorders [65,66]. Symptoms may be the main feature of the disease (such as ophthalmoplegia and ptosis in CPEO), or specific for a syndrome, such as optic nerve disease in Leber’s hereditary optic neuropathy (LHON). Otherwise, patients may manifest nonspecific ophthalmologic manifestations like cataracts, retinal disease, nystagmus, strabismus, or reduced visual acuity [27].

Mitochondrial damage is also involved in responsible mechanisms for retinopathy in patients affected by metabolic disorders (i.e., diabetes, dyslipidemia) and in several age-related retinal diseases, such as glaucoma and age-related macular degeneration. In fact, oxidative stress affects cell structures and components, particularly metabolically active neuronal cells such as photoreceptors. Therapies targeting these pathological processes were and are object of several studies in the literature [67,68,69,70,71].

Surgery may be beneficial for strabismus or cataracts and represent a valuable option for ptosis. The lubrification of the eyes is also important in patients who show inappropriate eyelid spread of tears in patients, with ptosis or after ptosis repair. 

LHON is a common mitochondrial optic neuropathy characterized by acute onset of painless and bilateral central vision loss. This disease more frequently develops in young adults. Risk factors such as heavy alcohol and moderate nicotine/tobacco exposure increase the possibility of developing symptoms and should be avoided. Idebenone, a short-chain benzoquinone capable of acting as an antioxidant agent, represents the only disease-specific drug approved to treat visual impairment in adolescents and adults with Leber’s hereditary optic neuropathy. Different trials have demonstrated the safety of this supplement in LHON treatment, although further investigations about its effectiveness are needed considering the rare incidence of this disease and the low recruitment numbers in clinical trials [9]. 

Several other therapies with neuroprotective, antioxidant, anti-apoptotic and/or anti-inflammatory activities have been investigated in the treatment of LHON but there are not clear results in clinical trials. Promising preliminary results have been reported for gene therapy in in-vitro and in-vivo models of LHON. In particular, allotopic gene therapy for LHON at low and medium doses appears safe, opening the door for testing at high doses [72].

### 4.5. Parkinsonism and Movement Disorders

Primary mitochondrial disease may be characterized by damage to basal ganglia, the cerebellum, the cortex, or corticospinal tracts, and may cause movement disorders and abnormalities of tone. Thus, patients can manifest a mixed movement and tone disorder, including hyper- and hypokinetic or cerebellar types of movements, hypotonia, spasticity, rigidity, and dystonia [25]. Other possible symptoms are myoclonus, ataxia, gait disturbance, parkinsonism, and rigidity. In the context of mitochondrial disorders, Leigh syndrome and Leber’s hereditary optic neuropathy mutations are often associated with dystonia, although this clinical manifestation has also been observed for several other mtDNA mutations. In particular, subjects with *POLG* mutations or mtDNA depletion syndromes are at higher risk of parkinsonian symptoms [73].

The clinical approach to movement and tone disorders in mitochondrial disease is comparable to that in patients with movement disorders of other causes. The symptomatic benefit of levodopa in mitochondrial patients with generalized dystonia or parkinsonism needs to be investigated further, as it is still not clear how pathophysiologic mechanisms involved in mitochondrial parkinsonism differ from those of parkinsonism of other causes. Symptoms like focal or multifocal dystonia can be improved with botulinum neurotoxin injections, whereas the use of oral baclofen, even if tolerated in most patients with generalized dystonia, has not provided satisfactory clinical responses [25]. 

Deep brain stimulation may be an option in the treatment of mitochondrial movement disorders, rigidity, and dystonia, considering the patient’s long-term prognosis and level of morbidity [27].

### 4.6. Anaesthesia

Patients with MDs may require general anesthesia in their diagnostic workup and subsequent management. However, the evidence base for the use of general anesthesia in these patients is limited and inconclusive. 

Mitochondria are a possible site of action for general and local anesthetics. Due to the high energy requirements of the central nervous system, patients with mitochondrial dysfunction might be more susceptible to changes in consciousness levels when anesthesia is used [74]. 

Patients affected by MDs are more susceptible to developing lactic acidosis, and surgical procedures and perioperative fasting, as possible metabolic stress factors, may exacerbated this condition. Routine, perioperative use of lactate-free intravenous fluids (such as 5% dextrose–0.9% saline) is recommended in all patients with MDs undergoing general anesthesia. 

The experts involved in the international Delphi-based consensus agreed that general anesthetic in MD patients was safe and that the risk of adverse events was exceptionally rare [14]. As already mentioned, in patients who are not on ketogenic diet, the perioperative administration of intravenous glucose associated with minimal preoperative fasting may prevent the effects of catabolism [14].

Regional anesthesia eliminates the risk of prolonged muscle relaxation and central nervous system depression, as well as the possibility of malignant hyperthermia. Although no neurological sequelae have been reported following spinal or epidural anesthesia, these procedures should be avoided in patients with major abnormalities of spinal cord or severe peripheral neuropathy [75]. 

## 5. Conclusions

Because of the lack of disease-modifying therapies for most MDs, the management of patients affected is mainly focused on the treatment of clinical features that patients show and on the prevention of major complications. In order to improve this approach, further knowledge with regard to mitochondria-safe drugs commonly used in clinical practice is essential. Furthermore, when the administration of a mitochondrion-toxic drug is needed, physicians should perform careful clinical and laboratory follow-up to precociously recognize and treat possible side effects (e.g., rhabdomyolysis, lactic acidosis, and hepatic failure, among others) [76]. 

Administration of vitamins, coenzyme Q, or other types of supplements is also an essential part of the therapeutic management of patients. For example, supplementation with folic acid has proved to be useful to compensate for the brain folate deficiency, which is a very common physiopathological aspect of mitochondrial diseases, especially in Kearns-Sayre syndrome [35].

Recently, besides recent trials which evaluated the potential therapeutic effects of idebenone in LHON [9,10], there is also a great interest in the new possibilities offered by mitochondrial biogenesis [77]. Gene therapy is mainly focused on the prevention of mtDNA related syndromes through the elimination of mutated mtDNA from oocytes [78], and the systemic or local administration of vectors to restore mitochondrial functionality [79]. Further clinical investigations and a better understanding of the mechanisms that this new approach may offer are needed. However, mitochondrial gene therapy appears to be to be a valuable and promising strategy to treat MDs.

## Figures and Tables

**Table 1 jcm-10-00094-t001:** Recommendations for safety use of the main drugs and drug classes in mitochondrial patients.

Drug (Classes).	Safe to Use? (Recommendations)	References
Adrenocorticotropic hormone	Yes	[20,21]
Aminoglycosides	Contraindicated in homoplasmic m.1555A>G and m.1494C>T mutations	[18]
**Antiepileptic drugs (AEDs)**		
Gabapentin	Yes	[22]
Lacosamide	Yes	[23]
Lamotrigine	Yes	[22]
Levetiracetam (LEV)	Yes	[22]
Phenobarbital	Yes	[22]
Phenytoin	Yes	[23]
Topiramate	Yes	[22]
Valproic acid	Contraindicated in POLG * mutations	[19]
Vigabatrin	Yes	[22]
Zonisamide	Yes	[22]
**Antivirals**		
Didanosine	No: may cause neuropathy	[24]
Fialuridine	No: may cause neuropathy or myopathy and lactic acidosis	[24]
Lamivudine	No: may cause neuropathy	[24]
Stavudine	No: may cause neuropathy or myopathy and lactic acidosis	[24]
Zalcitabine	No: may cause neuropathy	[24]
Zidovudine	No: may cause myopathy	[24]
Baclofen	Yes	[25]
Benzodiadepine (i.e., midazolam)	Yes	[23]
Carbamazepine	Yes	[14]
Cisplatin	Caution: may precipitate the development of hearing loss	[26]
Corticosteroids	Caution: may exacerbate myopathy if chronically used	[27]
Dichloroacetate (DCA)	Caution: causes peripheral nerve toxicity in mitochondrial myopathy, encephalopathy, lactic acidosis, and stroke-like episodes (MELAS)	[28,29,30]
**Dietary supplements**		
(Phospho)Creatine	Yes	[31,32,33,34]
CoQ10 (ubiquinone and ubiquinol)	Yes	[33,34]
Folic acid	Yes	[35]
l-Arginine	Yes	[36]
l-Carnitine	Yes	[34]
Lipoic acid	Yes	[29,37]
Riboflavin (vit. B2)	Yes	[38,39,40,41]
Enalapril	Yes	[14]
General anaesthetic (i.e., propofol)	Generally considered safe	[23]
Haloperidol	Yes	[14]
Idebenone	Yes	[7,8]
Metformin	Yes	[27]
Neuromuscular blocking agents	Not recommended in patients with myopathic phenotypes	
Oxcarbazepine	Yes	[14]
Paracetamol	Yes	[14]
Pyrimidine deoxynucleoside and deoxynucleotides	Yes	[42]
Statins	Caution: polymorphisms of uridine 5′-diphospho-glucuronosyltransferases (UGTs) and CIII could be predisposing factors in statin-induced myopathies	[43]
Elamipretide (experimental)	Yes	[44,45]
Omaveloxolone (experimental)	Yes	[46]

* POLG gene is located on chromosome 15 (15q26.1) and encodes DNA polymerase subunit gamma (POLG or POLG1). Pathogenic variants in POLG are the most common single gene causes of inherited mitochondrial disorders.

## Data Availability

No new data were created or analyzed in this study. Data sharing is not applicable to this article.

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
