# Peer review of "Therapeutical Management and Drug Safety in Mitochondrial Diseases—Update 2020"

_jcm, 2020, doi:10.3390/jcm10010094_

Round 1

Reviewer 1 Report

This is an important review providing updates re drug use in mitochondrial patients. I have only minor comments

1-Please clarify in the abstract if this review is about drugs specifically aimed to treat mitochondrial diseases, supplements/cocktails or any drug ie. AEDs , antibiotics etc. or both

2-Additionally to drugs what a about diets- ketogenic diet etc?

3-Row 54- considering mentioning also viral therapy for LHON, CoQ treatment for CoQ synthesis defects

4- Please check English language; For example oxidative phosphorylation chain: should be “oxidative phosphorylation” or “mitochondrial respiratory chain”

5- Consider adding article PMID: 19671450, which could provide additional info.

6- Summarizing the main point of tables would be nice; one for drugs to be avoided and one for possible treatments

Author Response

Dear Reviewer 1,
Thank you very much for your interest in our study; your comments were very helpful.
We here provide a point-by-point reply to them:

1-Please clarify in the abstract if this review is about drugs specifically aimed to treat mitochondrial diseases, supplements/cocktails or any drug ie. AEDs, antibiotics etc. or both
This has been specified (rows 18-19)
2-Additionally to drugs what a about diets- ketogenic diet etc?
This aspect has been briefly discussed (rows 81-85)
3-Row 54- considering mentioning also viral therapy for LHON, CoQ treatment for CoQ synthesis defects
This aspect has been briefly discussed (rows 74-77)
4- Please check English language; For example oxidative phosphorylation chain: should be “oxidative phosphorylation” or “mitochondrial respiratory chain” ---
English language has been revised
5- Consider adding article PMID: 19671450, which could provide additional info
This important study has been added in the text and in the reference list
6- Summarizing the main point of tables would be nice; one for drugs to be avoided and one for possible treatments
A table has been added, as suggested

Reviewer 2 Report

This review by Gruosso et al. is an excellent update on medication availability and efficacy for mitochondrial disorders. It is well organized and very comprehensive. It is recommended for publication in the Journal of Clinical Medicine following a revision that addresses the following concerns:

  1. A list of abbreviations is missing. It will improve readability of the text that relies heavily on acronyms.
  2. A table listing the drugs discussed in Section 2 should be provided. This is the main thrust of the review that should be presented in a table form.

Minor points:

line 92: pannelistis should be panellists 

line 321: involvment should be involvement

Author Response

Dear Reviewer 2,
Thank you very much for your interest in our study; your comments were very helpful.
We here provide a point-by-point reply to them:

1. A list of abbreviations is missing. It will improve readability of the text that relies heavily on acronyms.
We have now provided the list of abbreviations (Appendix)

2. A table listing the drugs discussed in Section 2 should be provided. This is the main thrust of the review that should be presented in a table form.
A table has been added, as suggested
Moreover, English language has been revised

I hope you a can find the revised manuscript (changes in bold) acceptable for publication.

Thank you very much and best regards

Michelangelo Mancuso

Reviewer 3 Report

JCM-1052123: Mitochondrial disorders and drugs - update 2020

In the current manuscript, authors have discussed the potential significance of mitochondrial homeostasis in the pathophysiology of disease majorly in neuropathy/retinopathy and discussed potential treatment candidates. The article is well constructed and elaborated the pathophysiology, potential role of mitochondrial dysfunction, and clinical outcome and discussed major pros and cons of current treatment.  I have a few suggestions mentioned below.

There is a need to modify the title of the manuscript relevant to the content and disease discussed in the text.

Mitochondrial disease associated with the pathogenesis of other diseases, listed in the current manuscript has already been extensively reviewed from time to time. The authors need to stress the merits of this review in the introduction and emphasize the need for this review article and discuss the current need for medicine relating to the disease listed here.  

It would be interesting to discuss the effects of dietary supplements on primary mitochondrial disorders such as highlight mitochondrial Folate (folic acid) metabolism and describing its role in mitochondrial disorders in listed diseases (PMID: 26183769, PMID: 26183769).

Alzheimer's disease (AD) is a major neurodegenerative disorder related to mitochondrial dysfunction. I would suggest if the author can discuss the current development of AD treatment by targeting mitochondria (PMID: 2166249).    

I recommend if the authors can depict the summary in one or two figures.

Line 90: author can list the advantages and disadvantages of 46 drugs that are reported to be safe in MD.

Line 147 needs to be rephrased according to the underline discussion.  

Line 279 3.3. Write the full name of PMM.

Line 417: The authors need to discuss key factors of glucose/ lipid metabolism in mitochondria responsible for retinopathy.    

Author Response

Dear Reviewer 3, thank you very much for your interest in our study. Your comments  were very helpful.
We here provide a point-by-point reply to them:

1. There is a need to modify the title of the manuscript relevant to the content and disease discussed in the text.
The title of the review has been changed in “Therapeutical management and safety drugs in mitochondrial disease – update 2020”
2. Mitochondrial disease associated with the pathogenesis of other diseases, listed in the current manuscript has already been extensively reviewed from time to time. The authors need to stress the merits of this review in the introduction and emphasize the need for this review article and discuss the current need for medicine relating to the disease listed here.
This aspect has been briefly discussed (rows 86-91)
3. It would be interesting to discuss the effects of dietary supplements on primary mitochondrial disorders such as highlight mitochondrial Folate (folic acid) metabolism and describing its role in mitochondrial disorders in listed diseases (PMID: 26183769, PMID: 26183769).
This aspect has been briefly discussed (rows 2911-4)
4. Alzheimer's disease (AD) is a major neurodegenerative disorder related to mitochondrial dysfunction. I would suggest if the author can discuss the current development of AD treatment by targeting mitochondria (PMID: 2166249).
This aspect has been briefly discussed (rows 63-67)
5. I recommend if the authors can depict the summary in one or two figures.---- A table has been added, as suggested
6. Line 90: author can list the advantages and disadvantages of 46 drugs that are reported to be safe in MD.
These aspects are now covered in the new table
7. Line 147 needs to be rephrased according to the underline discussion.
The line has been rephrased
8. Line 294 3.3. Write the full name of PMM. ---- We have written the full name of PMM
9. Line 417: The authors need to discuss key factors of glucose/ lipid metabolism in mitochondria responsible for retinopathy
These aspects have been briefly discussed (rows 2587-2592-67)

I hope you can find the revised manuscript (changes in bold) acceptable for publication.

Thank you very much and best regards

Michelangelo Mancuso

Round 2

Reviewer 3 Report

I carefully checked the manuscript. Authors have incorporated previous suggestions.
It looks decent and scientifically sound.
I feel that the manuscript needs sentence punctuation check!
Such as line 25, 26 and 29 can be merged in one paragraph.
Line 116-132 can be in one or two paragraphs.
Please ask to authors for these corrections.

Author Response

Dear Reviewer, thank you for your positive feedback. We have now revised the sentence punctuation, and also corrected few more typos around the text.

I hope you can now find this version of the manuscript acceptable for publication.

Best regards

Michelangelo Mancuso